# Differential Cytotoxicity of Curcumin-Loaded Micelles on Human Tumor and Stromal Cells

**DOI:** 10.3390/ijms232012362

**Published:** 2022-10-15

**Authors:** Xuan-Hai Do, My Hanh Thi Hoang, Anh-Tuan Vu, Lai-Thanh Nguyen, Dung Thi Thuy Bui, Duy-Thanh Dinh, Xuan-Hung Nguyen, Uyen Thi Trang Than, Hien Thi Mai, Thuy Thanh To, Tra Ngoc Huong Nguyen, Nhung Thi My Hoang

**Affiliations:** 1Department of Practical and Experimental Surgery, Vietnam Military Medical University, 160 Phung Hung Street, Phuc La, Ha Dong, Hanoi 10000, Vietnam; 2Faculty of Biology, VNU University of Science, 334 Nguyen Trai Street, Hanoi 10000, Vietnam or; 3Laboratory for Organogenesis and Regeneration, GIGA-R, University of Liège, 4000 Liège, Belgium; 4Center of Applied Sciences, Regenerative Medicine and Advance Technologies (CARA), Vinmec Healthcare System, 458 Minh Khai Street, Hanoi 10000, Vietnam; 5College of Health Sciences, Vin University, Hanoi 10000, Vietnam; 6Department of Biology, Mount Holyoke College, 50 College Street, South Hadley, MA 01075, USA

**Keywords:** breast cancer, curcumin, micelle, tumor spheroids, zebrafish, nano-cytotoxicity

## Abstract

Although curcumin in the form of nanoparticles has been demonstrated as a potential anti-tumor compound, the impact of curcumin and nanocurcumin in vitro on normal cells and in vivo in animal models is largely unknown. This study evaluated the toxicity of curcumin-loaded micelles in vitro and in vivo on several tumor cell lines, primary stromal cells, and zebrafish embryos. Breast tumor cell line (MCF7) and stromal cells (human umbilical cord vein endothelial cells, human fibroblasts, and human umbilical cord-derived mesenchymal stem cells) were used in this study. A zebrafish embryotoxicity (FET) assay was conducted following the Organisation for Economic Co-operation and Development (OECD) Test 236. Compared to free curcumin, curcumin PM showed higher cytotoxicity to MCF7 cells in both monolayer culture and multicellular tumor spheroids. The curcumin-loaded micelles efficiently penetrated the MCF7 spheroids and induced apoptosis. The nanocurcumin reduced the viability and disturbed the function of stromal cells by suppressing cell migration and tube formation. The micelles demonstrated toxicity to the development of zebrafish embryos. Curcumin-loaded micelles demonstrated toxicity to both tumor and normal primary stromal cells and zebrafish embryos, indicating that the use of nanocurcumin in cancer treatment should be carefully investigated and controlled.

## 1. Introduction

Breast cancer is one of the most common cancers for women worldwide and the second-leading cause of cancer-related death among females [1]. The high degree of heterogeneity of breast cancer makes it both a fascinating and challenging solid tumor to diagnose and treat [2,3]. It should be noted that normal breast development is controlled by the balancing of cell proliferation and apoptosis [4]. This balance is crucial in determining the overall growth or regression of the tumor in response to chemotherapy, radiotherapy, and, more recently, hormonal treatments [5,6,7]. Curcumin, a polyphenol derived from *Curcuma longa,* induces apoptosis by activating the mitochondrial apoptotic pathway [8]. This compound induces a loss of mitochondrial membrane potential, consequently opening the transition pore, releasing cytochrome c, activating caspase-9 and caspase-3, and cleaving PARP, ultimately leading to DNA fragmentation and apoptosis [6,8]. The down-regulation of anti-apoptotic proteins (Bcl-2 and Bcl-XL) and up-regulation of pro-apoptotic proteins (Bad and Bax) are also involved in curcumin-induced apoptosis in many cancer cell types, including breast cancer cells [8,9,10,11]. However, the high lipophilic nature of curcumin makes it poorly soluble in water, limiting its absorption through the gastrointestinal tract [12]. Therefore, the design of small and soluble curcumin is crucial for its pharmaceutical applications. Many studies have reported the synthesis and anti-tumor characteristics of different types of nanocurcumin [13,14,15]. These nanoparticles have exhibited multiple anti-tumor assets, such as drug labeling and tracking [16,17], apoptosis induction [8,18,19], and angiogenesis inhibition [20,21,22]. However, other recent studies considered curcumin one of the pan-assay interference compounds (PAINS), warning about its indiscriminate effects on normal cells [23,24,25]. Nanocurcumin, designed to enhance the therapeutic effect of curcumin, may exhibit a similar adverse impact on normal cells. Therefore, there is concern about the current use of different curcumin nanoformulas as dietary supplements for healthy individuals or cancer patients, without testing their bioactivity and, especially, safety.

Mesenchymal stem cells (MSCs), fibroblasts, and endothelial cells are the main players in tissue regeneration. To maintain cellular homeostasis in the organism, MSCs are activated, divided, and differentiated into the tissue-corresponding cells in damaged tissue to alleviate the injury [26,27]. Fibroblasts make up most of the stromal cells in all tissues and contribute to the production of the extracellular matrix by secreting collagen molecules, proteoglycans, and other substances. They are also found in connective tissues, such as bone, cartilage, and the skin’s loose connective tissue. As a result, fibroblasts play an essential role in regenerating injured tissues [28]. In tissue regeneration, angiogenesis is a crucial step that is primarily driven by the endothelial cells. These cells secrete angiocrine molecules and generate tissue-specific vascular subpopulations [29]. Whereas all three cell types could be sensitive to chemotherapy during cancer treatment [30], anti-cancer drugs should selectively destroy the cancer cells rather than the stromal cells.

In this study, we evaluated the bioactivity and toxicity of water-soluble curcumin-loaded copolymer micelles of hydrophilic poly(γ-glutamic acid) (γ-PGA) and α-tocopherol (Vitamin E) (curcumin PM), both in vitro using human cells and in vivo using zebrafish embryos. Our in vitro experiments focused on the anti-tumor effects on the breast cancer cell line MCF7, as well as its adverse effects on three non-cancerous primary stromal cell types, including umbilical cord vein endothelial cells (hUVECs), fibroblasts (hFBs), and umbilical cord-derived mesenchymal stem cells (UCMSCs). In addition, the in vivo effect of curcumin PM on the development of zebrafish embryos was also determined.

## 2. Results

### 2.1. Curcumin-Loaded Micelles Enhances Intracellular Uptake and Cytotoxicity

There is a correlation between the efficient accumulation of curcumin and its anti-cancer activity. To assess the intracellular uptake of curcumin and curcumin PM in two-dimensional (2D) cell culture monolayers, MCF7 cells and hUVECs were incubated with 30 µg/mL of either curcumin or nanocurcumin for 1 h. Fluorescence microscopy analysis revealed that the curcumin did not penetrate the cells; however, a strong and increasing green fluorescence signal was observed inside the nanocurcumin-treated cells over time (Figure 1A). The data indicated that the curcumin nanoformulation enhanced cellular uptake in comparison to free curcumin.

To examine the cytotoxicity, free curcumin and nanocurcumin at various concentrations were exposed to MCF7 cells, hUVECs, hFBs, and UCMSCs for 72 h. We observed no cytotoxic effect of free curcumin on the cancer cells and stromal cells at all tested doses (Figure 1B). In contrast, curcumin PM significantly inhibited the growth of MCF7 cells in a dose-dependent manner, with an IC_50_ value of 13.9 ± 0.5 µg/mL (Table 1). This curcumin PM-induced cytotoxicity was lower in hUVECs, hFBs, and UCMSCs with IC_50_ values of more than 20 µg/mL (Table 1). These results indicated that curcumin-loaded micelles inhibited the growth of cancer cells much more strongly than normal curcumin when dissolved in water. Moreover, curcumin PM showed more selective cytotoxicity toward breast cancer cells compared to stromal cells.

### 2.2. Curcumin-Loaded Micelles Infiltrated and Attenuated the Growth of MCF7 Spheroids

To characterize the therapeutic effect of curcumin in the tumor microenvironment, we evaluated the penetration and cytotoxicity of nanocurcumin in breast cancer spheroids. On day 5 of three-dimensional (3D) MCF7 cell culture, 30 µg/mL of free curcumin or nanocurcumin was added and quantitatively assessed under fluorescence microscopy at 4 h and 24 h. We detected strong fluorescence signals in the nanocurcumin-treated spheroids, whereas no signal was observed in free curcumin-treated samples after 4 h of incubation (Figure 2). Compared with that of curcumin, nanocurcumin uptake into the spheroids was faster and stronger in a time-dependent manner. At 4 h of incubation, the fluorescence signal was detected in a layer depth of 88µm; meanwhile, at 24 h, the signal was presented at 144 µm depth. Only a small amount of free curcumin could successfully penetrate the spheroids after 24 h (Figure 2). The results indicate that nanocurcumin had significantly higher spheroid internalization ability than curcumin.

To evaluate the cytotoxicity of curcumin on breast tumor spheroids, homogenous tumor spheroids were treated with the same dosages of free curcumin and curcumin PM and were measured as a function of time. The time-dependent changes in the morphology and diameter of the spheroids treated with either curcumin or nanocurcumin were measured over a 5-day time course (Figure 3). From day 3 of treatment, in the curcumin PM-treated spheroids, the outer layer cells became more loosely packed at all tested doses; meanwhile, this phenomenon was only seen at the highest dose of free curcumin (100 µg/mL) (Figure 3A). Moreover, the volume of curcumin PM-treated spheroids decreased in a dose- and time-dependent manner. The free curcumin-treated spheroids showed no significant change in the size and morphology, indicating the negligible effect of free curcumin at concentrations of 3, 10, and 30 µg/mL on the growth of 3D-cultured MCF7 cells, even after seven days of incubation (*p* > 0.05). However, at a higher concentration of 100 µg/mL, free curcumin inhibited the growth of spheroids, decreasing by 1.6 times, compared to the control (Figure 3B), after seven days (*p <* 0.05). On the other hand, the diameter of nanocurcumin-treated spheroids decreased, even at the lowest nanocurcumin concentration of 3 µg/mL, compared to the control (*p <* 0.01). The growth of spheroids was entirely inhibited when treated with 10, 30, or 100 µg/mL nanocurcumin for seven days (*p <* 0.0001). Taken together, we showed that nanocurcumin had a strong penetration ability and cytotoxicity toward breast cancer cells that were cultured in both monolayer and multicellular spheroid conditions.

### 2.3. Curcumin-Loaded Micelles Induced Apoptosis

Because nanocurcumin infiltrated and inhibited the growth of breast cancer cells in both 2D and 3D cultures, we next studied the ability of curcumin PM to induce apoptosis in hUVECs and MCF7 cells, using an annexin V staining assay. We found that nanocurcumin significantly triggered cellular apoptosis, compared to the mock control and free-curcumin-treated cells, especially at the higher dose of 30 µg/mL (32 ± 4 vs. 1.2 ± 0.1% in MCF7; 22.7 ± 1.2 vs. 2.3 ± 1.9% in hUVECs) (*p* < 0.05). The MCF7 cells showed increased sensitivity to the nanocurcumin-mediated apoptosis compared to hUVECs (*p* < 0.05) (Figure 4A,B). We further examined the curcumin-mediated apoptosis in the MCF7 cells using MitoTracker Red staining. After 24 h of incubation, the MitoTracker Red was shown as dense fluorescent dots, equally distributed at the perinuclear region in the control cells (Figure 4C). However, the fluorescence signals were either decreased or aggregated in the treated samples. In particular, nanocurcumin-treated cells aggregated at one side of cell nuclei after 24 and were then expressed in the nuclei after 48 h of treatment. In the meantime, the mitochondria did not appear as separated dots as in the control. In summary, the MitoTracker staining showed that the mitochondria changed in morphology and localization in cells treated with curcumin PM. These data suggest that nanocurcumin might induce MCF7 cell apoptosis via the mitochondria-mediated pathway.

### 2.4. Curcumin PM Attenuated the Tube Formation of hUVECs

Because the hUVEC viability was inhibited at an IC_50_ of 36.3 µg/mL (Table 1), lower concentrations (10 and 30 µg/mL) were used to evaluate the functional impact of curcumin PM on this cell type. Suramin, which was provided with the kit, was used as the positive control. The control and free curcumin-treated hUVECs on Matrigel formed the highest total tube length and tube-branching length, as well as tube junction number; meanwhile, Suramin showed the least (Figure 5A). In contrast, only a few, small tubes were found in the samples treated with 30 µg/mL of curcumin PM (Figure 5A). Quantitatively, nanocurcumin at concentrations of 30 µg/mL and 10 µg/mL suppressed the tube formation by nearly 100% and 81%, respectively (Figure 5B). These results suggest the angiogenesis inhibitory function of curcumin PM in vitro.

### 2.5. Curcumin PM Prevented the Migration of hFBs and UCMSCs

Cell migration plays an essential role in fibroblasts and UCMSCs-mediated tissue repair. Hence, we performed a wound-healing assay to assess the effect of curcumin PM on this process. We observed that nanocurcumin significantly inhibited the migration of both hFBs and UCMSCs at two doses of 30 and 10 µg/mL (*p* < 0.01) (Figure 6A). In addition, the compound demonstrated a higher inhibitory effect on mesenchymal stem cells compared to fibroblasts, with a reduction in cell migration even at 3 µg/mL (*p* < 0.05) (Figure 6B). After 24 h, curcumin PM at a concentration of 30 µg/mL mediated the wound covering of 6.3% and 26.6% in hUVECs and hFBs, respectively, compared to almost 100% recovery of the controls, indicating the inhibitory effect of curcumin PM on stromal cell movement.

### 2.6. Toxicity of Curcumin-Loaded Micelles in Zebrafish Embryos

We evaluated the in vivo toxicity of nanocurcumin on zebrafish embryos by measuring the survival and malformation rate (%) of the curcumin-treated embryos in a dose and time-dependent manner (Figure 7A). The embryos’ survival was judged by the presence of heartbeats and the clarity of the embryos. There was no significant correlation between the incubation time and survival rates (Table 2). Meanwhile, the malformation rates were significantly increased between 24 and 96 h of exposure to the substances, with EC50 values ranging from 18.1 to 57.4 mg/L (Table 2, *p* < 0.05). At 96 h, the teratogenic index (TI, defined as the ratio between LC_50_ and EC_50_) of curcumin PM was ~1.38, indicating weak teratogenicity. The embryonic malformation was assessed based on these categories: pericardial edema, heart malformation, lordosis, yolk sac edema, swim bladder un-inflation, tail malformation, abnormal tail length, yolk necrosis, decreased pigment in eyes or body, and congestion of the peripheral blood circulation. In our experiment, the obtained malformation types were undeveloped embryos, a yolk sac edema, yolk necrosis, congestion of peripheral blood circulation, and a hook-like tail (Figure 7B).

## 3. Discussion

Curcumin is well known as an effective antioxidant, anti-inflammatory, and anti-tumor agent. However, the hydrophobicity of free curcumin inhibits its clinical application [9]. In our study, nanocurcumin with a water solubility of 10% expressed higher effects in inhibiting the growth of the breast cancer cell line MCF7 and induced cellular apoptosis, compared to free curcumin. In the culture medium, free curcumin aggregated in the solution (Figure 2), preventing its penetration into cells and decreasing its effects. In accordance with previous reports, the nano formulation of curcumin facilitates its cellular uptake, thereby expressing higher cytotoxicity and inducing breast cancer cell apoptosis [12,14]. The anticancer activity of curcumin copolymeric micelles in this current study (with an IC_50_ value of 13.9 ± 0.5 µg/mL) was even lower compared to the values reported in the previous publications by Hosseini et al., with an IC_50_ for the nano-micelle curcumin of 59.72 mmol/L on breast cancer cells [31]. 

In the drug screening, a number of substances showed a high effect on in vitro monolayer cell culture but failed to exert an in vivo effect [32,33]. This was because the multiple-layer nature of cells in the tumor hampered the tumor penetration of many in vitro-effective compounds [34,35,36]. Therefore, we used the multicellular tumor spheroid model to evaluate the therapeutic effect of curcumin. We observed that nanocurcumin could infiltrate MCF7 cells that were cultured as both monolayer and multicellular tumor spheroids. Using z-stack imaging, we found that the nanocurcumin presented strongly in several cell layers, even at a depth of 140 µm from the top of the spheroid, which corresponded to about 10 layers of cells. Meanwhile, the fluorescence signal of the free curcumin was only observed at 72 µm depth. This could explain why nanocurcumin demonstrated a stronger effect in terms of preventing MTS growth from a dose of 10 µg/mL, compared to free curcumin. Even a lower concentration of 3 ug/mL of nanocurcumin could inhibit the growth rate of MCF7 spheroids, indicating that nanocurcumin could be a potential anti-cancer drug substrate.

To be a potential therapeutic drug, the drug substrate should simultaneously show therapeutic effects on targeted cells and safety profiles on normal cells. Therefore, we evaluated the impact of curcumin PM on the viability and function of human endothelial cells. Even though this nanocurcumin system was less toxic to hUVECs than to breast cancer cell MCF7, the compound showed strong angiogenesis inhibition activity. Our results were consistent with previous studies presenting curcumin as an angiogenesis inhibitor, contributing to its anti-tumor activity [20,21,22]. Because angiogenesis is a necessary process for tissue development and regeneration [37], inhibiting this process could be harmful to the body. In fact, we revealed that nanocurcumin also demonstrated cytotoxicity to human fibroblast cells, as well as to human mesenchymal stem cells. However, incubating these cells with nanocurcumin significantly reduced the migration of the cells. Since these stromal cells are key players in tissue repair, recovery, and regeneration [37,38,39], the negative impact of nanocurcumin on these cells should be carefully considered when developing curcumin as a drug substance.

Curcumin has been used as a spice, coloring agent, and folk medicine in many Asian countries and is considered a safe compound. Moreover, no toxicity was observed concerning fertility or pregnancy in curcumin-fed rats, and no malformation in their embryos [40]. However, some researchers recently reported that curcumin caused zebrafish embryo malformation at lower concentrations (<7.5 µM) and death at higher concentrations (>12.5 µM) [41]. The zebrafish is a standard model for evaluating potential teratogen invertebrates. Since several kinds of nanocurcumin have been used as a dietary supplement and could be developed as anti-cancer drugs, they should be tested for safety. This study found that nanocurcumin affected zebrafish embryonic development at a concentration as low as 15 mg/L, with a weak teratogenic potential. Previous studies reported that when incubating it with zebrafish embryos, curcumin could accumulate in the edema sac area and be randomly distributed on the larvae’s skin without excretion [34]. In the size of nanoparticles, nanocurcumin is absorbed much more easily into the embryos and larvae than free curcumin, thereby being lethal to the embryos. The abnormal development of zebrafish, when treated with nanocurcumin, may result from the inhibition of stromal cell migration and angiogenesis, as we discussed above. Our study was consistent with a recent study by Cao et al., showing that nanocurcumin inhibits angiogenesis via down-regulating hif1a/VEGF-A signaling in zebrafish [20].

Based on the results of this study, we supposed that nanocurcumin might be a good candidate for anti-cancer drug development. However, a therapeutically appropriate dosage should be carefully determined to balance safety and efficacy [42]. Combining nanocurcumin with other natural or synthetic products may also be considered to increase the benefits and minimize the unwanted risks [43]. Besides, nanocurcumin should not be used as a spice, food additive, or daily supplement food since this nanoform can negatively affect stromal cells.

## 4. Materials and Methods

### 4.1. Materials

Curcumin-loaded micelles were fabricated based on an amphiphilic graft polymer composed of hydrophilic poly(γ-glutamic acid) (γ-PGA) and α-tocopherol (Vitamin E). The size of micelles is from 50–70 nm (Figure 8), with a water solubility of 10%. This material was provided by Associate Professor Pham Huu Ly, Ph.D., from the Institute of Chemistry (Vietnam).

Free curcumin was purchased from Sigma-Aldrich (St. Louis, MO, USA). All other reagents and buffer solution components were of prepared analytical grades.

### 4.2. Cell Culture

MCF7 human breast cancer cells were obtained from ATCC and maintained in Dulbecco’s modified Eagle medium (DMEM, Gibco, New York, NY, USA) supplemented with 10% fetal bovine serum and 100 units/mL of penicillin, along with 100 μg/mL of streptomycin (Gibco, New York, NY, USA). The cells were cultured in a humidified incubator at 37 °C with 5% CO_2_. Cells were sub-cultivated every 3 or 4 days when the cultures reached 70–80% confluence.

Human fibroblasts (hFBs) were cultured in DMEM/F12 medium (Gibco, New York, NY, USA) supplemented with 10% fetal bovine serum (FBS) (Gibco, New York, NY, USA), 100 units/mL of penicillin, and 100 µg/mL of streptomycin (Gibco, New York, NY, USA). Human umbilical vein endothelial cells (hUVECs) were cultured in an EBM-2 medium kit (Lonza, Basel, Switzerland). Umbilical cord-derived mesenchymal stem cells (UC-MSCs) were grown on the culture flask surface coated by CELLstart™ CTS™ (CELLstart) in StemMACS^TM^ MSC expansion media (StemMACS) (Miltenyi Biotec, Bergisch Gladbach, Germany). All the cells of hUVECs, hFBs, and UC-MSCs were primarily isolated and incubated at 37 °C with 5% CO_2_.

### 4.3. Three-Dimensional Culture Cell Formation

The MCF7 spheroids were generated by a modified hanging drop, as previously described [34]. Briefly, 15 µL of medium containing 5 × 10^3^ cells were added to each circle on the inverted cover, plated on a 96-well plate for making one spheroid. The upside-down cover was transferred to an agarose-coated plate. After 48 h of incubation, the spheroids were transferred to agarose-coated wells for further cultivation. The morphology of spheroids was captured by an Axiovert 40CFL microscope (Zeiss, Oberkochen, Germany) with a Canon Powershot G9 camera (Canon, Japan). The diameter of spheroids was analyzed using Axio, version 4.5 (Zeiss, Oberkochen, Germany).

### 4.4. Cellular Uptake Study

MCF7 cells and hUVECs were seeded onto glass coverslips and placed in 24-well plates for 24 h in suitable media, as indicated above. Cells were then incubated with 30 µg/mL of either curcumin or curcumin-loaded micelles for 1 h. The concentration used in this experiment was chosen based on the previous papers [16,17], with a higher dose to follow in a shorter time of incubation to check the compound-uptake status of the cells. At the end of incubation, cells were rinsed and fixed with 4% paraformaldehyde and 2% sucrose at 37 °C. Coverslips were mounted on glass slides. The nuclei were stained with Hoechst 33342 (Invitrogen, Waltham, MA, USA). The MCF7 spheroids were cultured on an agar-coated 96-well plate and incubated with 10 and 30 µg/mL of either curcumin or curcumin-loaded micelles for 24 h. The uptake of curcumin, as well as nanocurcumin, was estimated via a fluorescence signal. Images were collected with a Zeiss 510 laser-scanning confocal microscope, with 40× or 63× objectives.

### 4.5. Cytotoxicity Assay

The cytotoxicity of curcumin and curcumin-loaded micelles against MCF7 cells and stromal cells was evaluated by using an in vitro MTT assay, which is based on the reduction of MTT by the mitochondrial dehydrogenase of intact cells to a purple formazan product. Cells were seeded into a 96-well plate in normal growth conditions. Different concentrations of curcumin (in both free and curcumin-loaded micelles) were prepared by adding the required amounts directly into the culture medium. The final concentrations ranged from 2.3 to 600 μg/mL. Each sample was prepared in triplicate. The same volume of medium was used for the control wells. At the end of the time point, 3-(4,5-dimethylthiazol-2-yl)-2,5-diphenyltetrazolium bromide (MTT) reagent was added to each well and used according to the manufacturer’s instructions (Promega). The optical density measurement was carried out at 570 nm, using a microplate reader (BioRad, Hercules, CA, USA). The IC_50_ value was defined as the sample concentration that reduced absorbance by 50%, compared to the control.

### 4.6. Multicellular Tumor Spheroid Assay

After landing on agar-coated wells in a 96-well plate, MCF7 spheroids were treated with free curcumin and nanocurcumin at concentrations of 3, 10, 30, and 100 µg/mL for 7 days. The range of concentrations used in this assay was based on the results of the cytotoxicity test. Images were obtained daily using an Axiovert 40CFL microscope (Carl Zeiss AG) with a Powershot G9 camera (Canon, Japan). These images were analyzed using the Axio software, version 4.5 (Carl Zeiss AG), to determine the spheroid diameter. The approximate volume (V) of each spheroid was calculated as follows: V = (4/3) × π v (D1/2) × (D2/2)^2^, where D1 and D2 were the longest and shortest diameters, respectively (26).

### 4.7. MitoTracker Assay

MCF7 cells and hUVECs were grown on coverslips for approximately 15 h. After complete adhesion, the cells were treated with either 10 µg/mL or 30 µg/mL free curcumin and curcumin-loaded micelles for the indicated time at 37 °C, for two time points of 24 h and 48 h (for MitoTracker Red staining) and 24 h (for Annexin V staining). Next, the cells were rinsed and live-stained with MitoTracker Red CMXRos or Annexin V-FITC (conjugated) (Invitrogen, Waltham, MA, USA) for 24 h in a serum-free medium for 30 min at 37 °C. Then, the culture medium was removed, and coverslips were washed thoroughly with PBS and mounted on glass slides. The images were analyzed using a confocal microscope, LSM 510 (Zeiss).

### 4.8. Wound-Healing Assay

Cells were cultured in different media, including DMEM/F12 (hFBs) and StemMACS (UCMSCs), and seeded on 24-well plates at a density of 8000 cells/cm^2^ until the cells presented >90% confluence on the plates. When the cells covered the culture plate at a density of >95% density, cells were treated with mitomycin (10 µg/mL) for 2 h to inhibit cell proliferation before creating the wound by using a scratcher (SLP, Korea). Afterward, the cells were supplemented with curcumin PM at concentrations of 0 (negative control), 3, 10, and 30 µg/mL. The process of wound healing was observed and captured by the optical microscope. The image results were analyzed with ImageJ software (version 1.46r).

### 4.9. Angiogenesis Assay

The cells were cultured in an EBM-2 medium until they reached 80% confluence and were then collected for the angiogenesis experiment, using an Angiogenesis Assay Kit (Abcam, Cambridge, UK) following the manufacturer’s protocol. The cells were seeded on 6-well plates at a density of 2 × 10^5^ cells/well, and all the wells were coated with an extracellular matrix solution supplied with the kit. The experimental cell wells were treated with free curcumin and nanocurcumin at concentrations of 10 µg/mL and 30 μg/mL in hUVECs, or 30 µg/mL in hFBs and UCMSCs. Suramin (50 μg/mL) was added to the negative control wells. The cells were incubated at 37 °C and 5% CO_2_ conditions for 10 h. The angiogenesis processes in the wells were observed and captured by fluorescence optical microscopy and electron microscopy using a filter with a wavelength of 490/540 nm. Image analysis was achieved using Image J software (version 1.46r).

### 4.10. Substance Treatment and Embryo Assessment

Zebrafish, *Danio rerio*, were purchased from local fish stores and were maintained in the laboratory at a temperature of 26 °C ± 1 °C and a 14 h day/10 h night photoperiod, in E3 medium. Those individuals presenting good fecundity were selected for breeding.

The method of assessing the toxicity of zebrafish embryos was conducted following OECD Test No. 236 [44]. After breeding, healthy embryos at 2 h post-fertilization (2-hpf) were individually placed in 24-well microtiter plates containing the test substance diluted in E3 medium (5 mM NaCl, 0.17 mM KCl, 0.4 mM CaCl_2_, and 0.16 mM MgSO_4_), then incubated at 26 °C. The tested concentrations were 0 (control), 10, 15, 20, 30, 45, and 60 mg/L. Every 24 h, the media were renewed, and the number of defective (including dead) and hatched embryos was recorded until 96-hpf. All experiments were conducted in triplicate on *n* = 20 embryos per concentration.

### 4.11. Statistical Analysis

All statistical tests and toxicological indices were calculated using GraphPad Prism 5. The differences between groups were assessed using an unpaired Student’s *t*-test, a two-way analysis of variance (ANOVA), and Tukey’s HSD tests. A *p*-value < 0.05 was considered to indicate a statistically significant difference. All data are presented as the mean ± SD.

## 5. Conclusions

Curcumin PM nanoparticles could inhibit the growth of MCF7 cells and tumor spheroids. However, nanocurcumins were cytotoxic to stromal cells, including endothelial cells, fibroblasts, and mesenchymal stem cells. Moreover, nanocurcumin was lethal and teratogenic to zebrafish embryos from a concentration of ≥ 15 mg/L. Therefore, we recommend that any nanocurcumin drug development should find reasonable solutions to prevent any side effects on normal cells—for instance, by exploring the therapeutical use of nanocurcumin in tandem with other natural or synthetic agents.

## Figures and Tables

**Figure 1 ijms-23-12362-f001:**
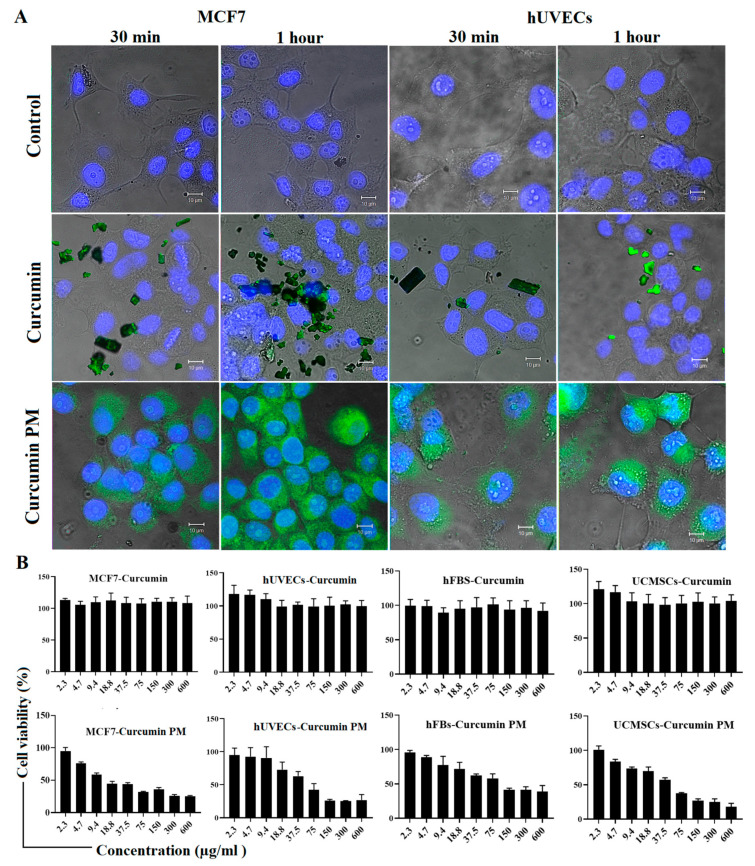
Intracellular uptake and cytotoxicity of curcumin and curcumin PM in stromal cells and MCF7 cells in a two-dimensional culture. (**A**) The internalization of curcumin molecules was observed in curcumin PM–treated cells but not in curcumin-treated ones after 30 min and 1 h of incubation at 30 µg/mL. (**B**) The cell viability (%) of stromal cells and MCF7 cells treated with curcumin and curcumin PM. The data are presented as mean ± SD, N = 3. Scale bar: 10 µm.

**Figure 2 ijms-23-12362-f002:**
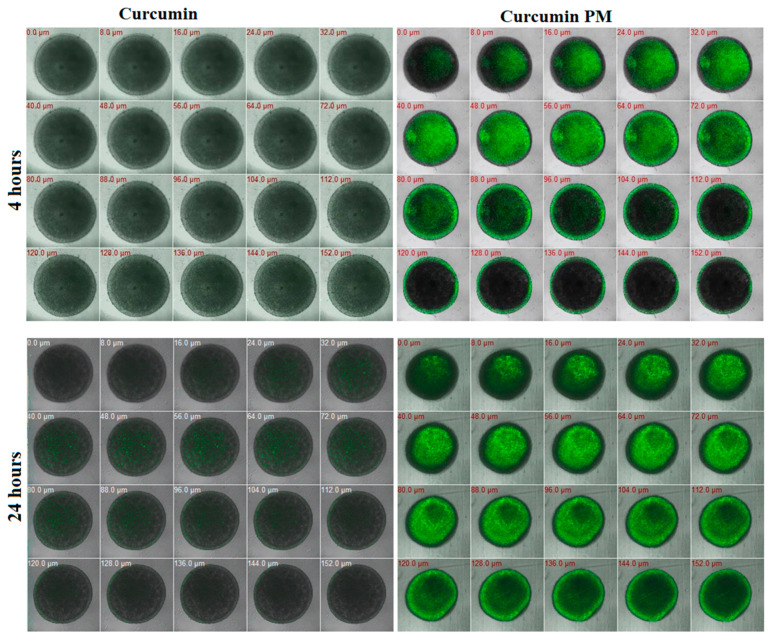
The absorption of curcumin PM into MCF7 spheroids was visualized by the autofluorescence of curcumin in the nanoparticle complex. Spheroids were incubated with nanocurcumin at a concentration of 30 µg/mL. The images were taken after 4 h and 24 h of incubation on LSM 510. The upper-left value in each image represents the depth from the spheroid’s top (i.e., 0.0 µm means that this is the spheroid’s top surface). The fluorescence level reached a maximum after 24 h of incubation. It can be seen that curcumin PM is present not only on the outside but also on the inside, near the necrotic core of spheroids; meanwhile, a very weak fluorescence signal was observed in the free curcumin.

**Figure 3 ijms-23-12362-f003:**
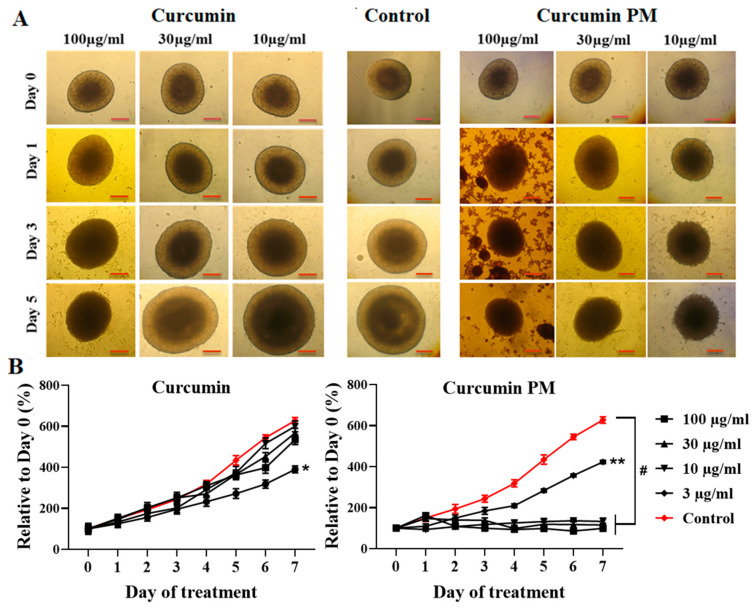
Curcumin PM successfully penetrated and inhibited the growth of MCF7 spheroids. (**A**) Morphology of spheroids of MCF7 cells incubated with either curcumin or nanocurcumin. The images were taken using an Axiovert 40CFL microscope (Zeiss, Germany) with a Canon Powershot G9 camera (Canon, Japan), and the diameters were analyzed by Axio version 4.5 (Zeiss, Germany). The scale bar represents 200 μm. (**B**) The growth curves of spheroids, when incubated with free curcumin and curcumin PM. The curves represented the dose-dependent behavior of MCF7 spheroids under two treatment conditions. The data are presented as mean ± SD, N = 3. * *p* < 0.05; ** *p* < 0.01; # *p* < 0.0001. Scale bar: 200 µm.

**Figure 4 ijms-23-12362-f004:**
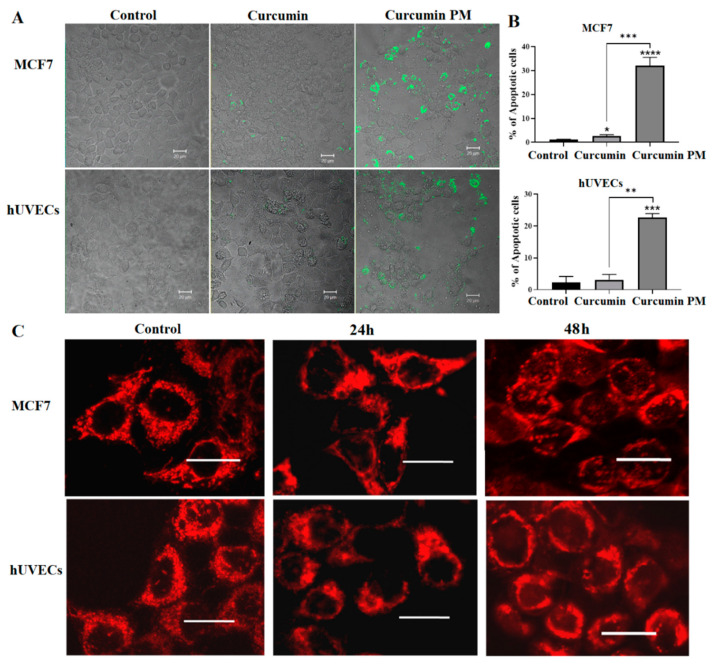
Curcumin-loaded micelles induced apoptosis in both hUVECs and MCF7 cells. (**A**) Annexin V staining revealed the expression of an apoptotic marker in the cells treated with curcumin PM. (**B**) The percentage of apoptotic cells after incubating with free-cucurmin and nanocurcumin at 30 µg/mL. (**C**) MitoTracker Red staining indicated the mislocalization of mitochondria in the treated cells. Treated hUVECs and MCF7 cells were incubated with MitoTracker Red (100 nM) for 30 min at 37 °C. The cells were washed with PBS and the live cells were imaged. The fluorescence signal was reduced in the samples treated with curcumin-loaded micelles, following 24 h and 48 h of incubation compared to the control. The data are presented as mean ± SD, *N* = 3. * *p* < 0.05; ** *p* < 0.01; *** *p* < 0.001; **** *p* < 0.0001. Scale bar: 10 μm.

**Figure 5 ijms-23-12362-f005:**
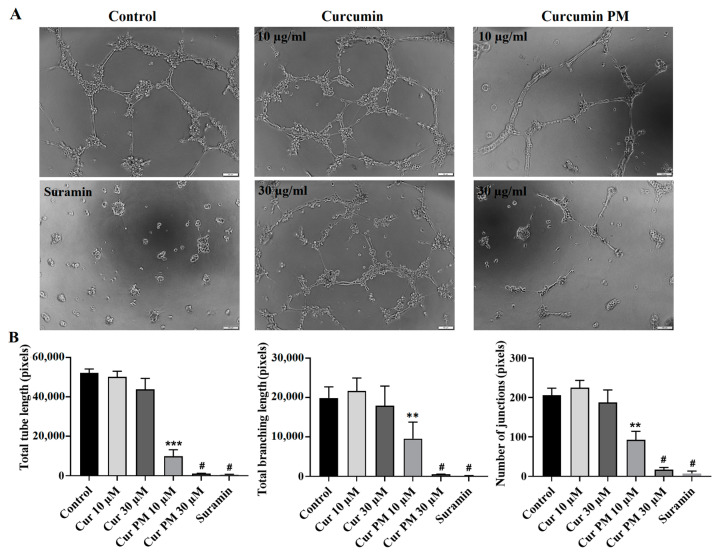
The effect of curcumin PM on the tube formation of hUVECs. (**A**) The morphology of tube-like structures in those cells incubated with free curcumin and curcumin PM at two concentrations of 10 and 30 µg/mL. (**B**) Quantitative analysis of tube formation ability in hUVECs under different culture conditions. The data are presented as mean ± SD, *N* = 3. ** *p* < 0.01; *** *p* < 0.001; # *p* < 0.0001. Cur: curcumin; Cur PM: curcumin PM. Scale bar: 50 µm.

**Figure 6 ijms-23-12362-f006:**
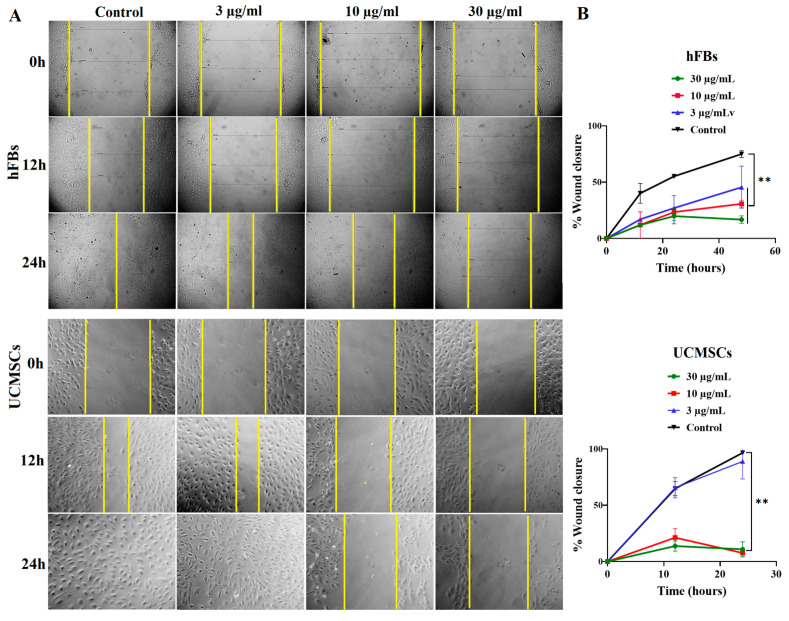
The effect of curcumin PM on the migration of UCMSCs and hFBs. (**A**) The images show wound closure (%) following a time of incubation with curcumin PM at two doses of 10 and 10 µg/mL. (**B**) The quantitative percentage of the wound covered after treatment with curcumin PM in UCMSCs and hFBs. The data are presented as mean ± SD, *N* = 3. ** *p* < 0.01.

**Figure 7 ijms-23-12362-f007:**
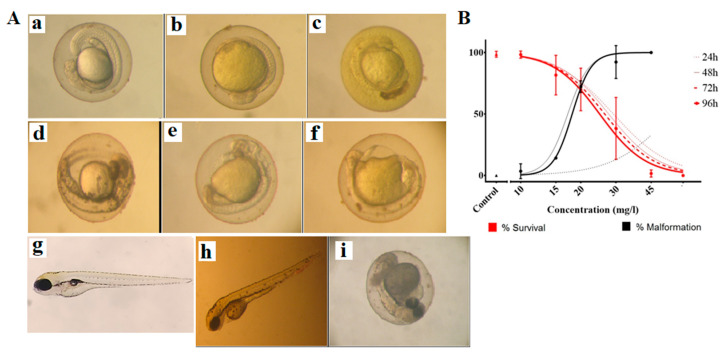
(**A**) Malformations in zebrafish embryos exposure to curcumin-loaded micelles. (**a**) Control at 24 h; (**b**) 24 h of exposure at 30 mg/mL, the yolk turned yellow without observable elongation; (**c**) 24 h of exposure at 45 mg/mL, yellow chorion and no elongation; (**d**) control at 48 h; (**e**) 48 h of exposure to 20 mg/mL, yellow yolk, no observable embryonic segmentation, no eye nor body pigmentation; (**f**) 48 h of exposure at 30 mg/mL, yellow and necrosed yolk, no embryonic segmentation nor pigmentation; (**g**) control at 96 hpf; (**h**) 96 h of exposure at 20 mg/mL, black and necrosed yolk, congested blood circulation in the tail; (**i**) After 96 h of exposure at 30 mg/mL, the larvae could not escape from the chorion, showing necrosis in the tail and yolk and pericardial edema. (**B**) The dose-response curve of nanocurcumin on zebrafish embryos. The data are presented as mean ± SD, N = 3.

**Figure 8 ijms-23-12362-f008:**
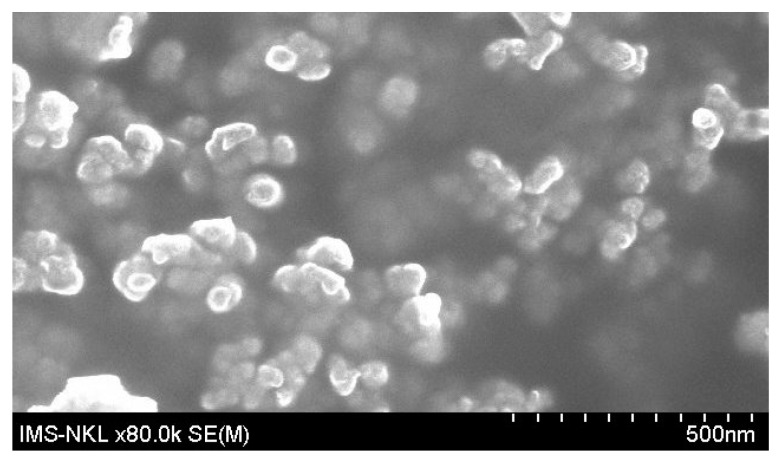
Field emission scanning electron microscopy (FeSEM) image of curcumin-loaded micelles.

**Table 1 ijms-23-12362-t001:** The IC_50_ values of free curcumin and curcumin PM on different cell types (ND: not not applicable).

	MCF7	hUVECs	hFBs	UCMSCs
Curcumin	NA	NA	NA	NA
Curcumin PM (µg/mL)	13.9 ± 0.5	36.96 ± 1.2	28.34 ± 2.4	26.62 ± 5.1

**Table 2 ijms-23-12362-t002:** Toxicity of curcumin-loaded micelles in zebrafish embryos.

Time of Exposure (h)	LC_50_ (mg/L)	EC_50_ (mg/L)
**24**	29.1	57.4
**48**	27.6	17.2
**72**	26.4	
**96**	24.9	18.1
	(*p* > 0.05)	(*p* < 0.05)

## Data Availability

Not applicable.

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
