# Peer review of "Differential Cytotoxicity of Curcumin-Loaded Micelles on Human Tumor and Stromal Cells"

_ijms, 2022, doi:10.3390/ijms232012362_

Round 1

Reviewer 1 Report

The paper presents the in vitro and in vivo toxicity of curcumin-loaded micelles (curcumin PM) on several tumor cell lines, primary stromal cells and zebrafish embryos.

In my opinion the manuscript contains enough original and interesting material. It is written clearly and concisely. The experimental procedures are described comprehensively. The results are interesting.

Minor corrections:

The authors should compare the anticancer activity of the curcumin PM with that of the reference anticancer drug.

Figure 7: there is no photo of zebrafish larvae from the control group at 96 hpf.

Line 43: Latin names should be written in italics - Curcuma longa.

Lines 231, 241, 295: there is: Zebrafish, should be: zebrafish.

Line 396: there is: Danio renio, should be: Danio rerio.

References should be adapted to the journal's requirements.

The language should be modified carefully.

Author Response

Responses to Reviewer’s comments and suggestions

Dear Editors and Reviewers,

We would like to thank you for your highly valuable comments and suggestions. Please find below our answers to your list of queries.

**********

Reviewer 1

  1. The authors should compare the anticancer activity of the curcumin PM with that of the reference anticancer drug.

We have added in the discussion section the comparison of the anticancer activity of curcumin PM with another nano-micelle curcumin published before: “The anticancer activity of curcumin copolymeric micelles in this current study (with an IC50 value of 13.9 ± 0.5 µg/ml) was even lower compared to that in the previous publications of Hosseini S et al. with an IC50 of nano-micelle curcumin of 59.72 mmol/L on breast cancer cells [31]”. (Lines 283-286). 

  1. Figure 7: there is no photo of zebrafish larvae from the control group at 96 hpf.

We have added the control group at 96 hpf, and revised Figure 7.

  1. Line 43: Latin names should be written in italics - Curcuma longa. We have corrected to italics name
  2. Lines 231, 241, 295: there is: Zebrafish, should be: zebrafish. We have corrected Zebrafish into zebrafish
  3. Line 396: there is: Danio renio, should be: Danio rerio. We have corrected Danio renio into Danio rerio
  4. References should be adapted to the journal's requirements. We have adjusted the references to adapted to the journal's requirements.
  5. The language should be modified carefully. We have thoroughly revised language using Grammarly

Sincerely yours,

Nhung HOANG

Reviewer 2 Report

1.       At line 43, authors should italicized the word “Curcuma longa”.

2.       From line 54-56 stating “These nanoparticles to … angiogenesis inhibitor”, authors are encouraged to restructure the sentence and end it appropriately.

3.       Across the study, authors should provide brief explanations for the range of concentration used in curcumin treatment (e.g. why 30 mg/mL is used for free curcumin and curcumin PM), as well as supportive citations if available.

4.       In general, figures can be improved.

a.       For Figure 1(A), please include the scale and ensure that the resolution is at least 300 dpi. The background of the fluorescence microscopy should appear to be darker.

b.       For Figure 3(A), some scales are also missing. It should be included for all sampling groups from Day 1, 3 and 5.

c.       For Figure 5(A), author did not explain about the role of suramin as positive control. Quantitative analysis for suramin tube formation should also be included in Figure 5(B).

5.       Line 252-253 requires citation and examples.

6.       At line 287, authors have to include the range tested.

7.       From line 296-299, authors’ conclusion might be too hasty and bias. Authors are highly recommended to include other supporting evidence. There is also concern regarding the dosage, as similar papers applying lower dosage in their curcumin-based nanoformulation, e.g. refer to the article “Free and nano encapsulated curcumin prevents scopolamine-induced cognitive impairment in adult zebrafish (Coradini et al. 2021)”.

8.       At line 416, authors should include examples for the mM concentration.

9.       In terms of conclusion, authors are required to reiterate important findings from the study. Authors may like to state the reasonable solutions (future studies) as to how to use curcumin PM as an anticancer agent. For example, it might be used in tandem with another natural product or synthetic drug agent.

Author Response

Responses to Reviewer’s comments and suggestions

Dear Editors and Reviewers,

We would like to thank you for your highly valuable comments and suggestions. Please find below our answers to your list of queries.

**********

Reviewer 2

  1. At line 43, authors should italicize the word “Curcuma longa”. We have corrected to italics name
  2. From line 54-56 stating “These nanoparticles to … angiogenesis inhibitor”, authors are encouraged to restructure the sentence and end it appropriately.

We have rewritten the sentence into “These nanoparticles have been demonstrated multiple anti-tumor assets, such as drug labelling and tracking [16, 17], apoptosis induction [8, 18, 19], and angiogenesis inhibitor [20-22]”. (Lines 57-59).

  1. Across the study, authors should provide brief explanations for the range of concentration used in curcumin treatment (e.g. why 30 mg/mL is used for free curcumin and curcumin PM), as well as supportive citations if available.

We have provided brief explanations for the range of concentrations used for curcumin in this study in the method section, such as “The concentration used in this experiment was chosen based on the previous papers [16,17] with a higher dose to follow in a shorter time of incubation to check the compound-uptake status of the cells”. (Lines 382-385); “The range of concentrations used in this assay was based on the results of the cytotoxicity test”. (Lines 409-410).

  1. For Figure 1(A), please include the scale and ensure that the resolution is at least 300 dpi. The background of the fluorescence microscopy should appear to be darker.

We have added the scale bar and darkened the background in the fluorescence images in Figure 1A. The revised figure was resubmitted.

  1. For Figure 3(A), some scales are also missing. It should be included for all sampling groups from Day 1, 3 and 5.

We have added a scale bar to all images from Day 0 to Day 5. The revised figure was resubmitted.

  1. For Figure 5(A), author did not explain about the role of suramin as a positive control. Quantitative analysis for suramin tube formation should also be included in Figure 5(B).

We have added the explanation about the role of Suramin as a positive control (Lines 200-201). Quantitative analysis for suramin tube formation was also included in Figure 5(B).

  1. Line 252-253 requires citation and examples

We have added the citation for this information. (Line 288).

  1. At line 287, authors have to include the range tested

We have included the range tested in this sentence. (Lines 302 – 303).

  1. From line 296-299, authors’ conclusion might be too hasty and bias. Authors are highly recommended to include other supporting evidence. There is also concern regarding the dosage, as similar papers applying lower dosage in their curcumin-based nanoformulation, e.g. refer to the article “Free and nano encapsulated curcumin prevents scopolamine-induced cognitive impairment in adult zebrafish (Coradini et al. 2021)”

We have rewritten this part to “Based on the results of this study, we supposed that nanocurcumin might be a good candidate for anti-cancer drug development. However, therapeutically appropriate dosage should be carefully determined to balance safety and efficacy [42]. The combination between nanocurcumin and other natural or synthetic products may also be considered to increase benefits and minimize unwanted risk [43]. Besides, nanocurcumin should not be used as a spice, food additive, or daily supplement food since this nanoform can negatively affect stromal cells”. (Lines 335-343).

  1. At line 416, authors should include examples for the mM concentration.

We have rewritten the sentence “Moreover, nanocurcumin was lethal and teratogenic to zebrafish embryos from the concentration of ≥ 15 mg/L” (Lines 464 – 465).

  1. In terms of conclusion, authors are required to reiterate important findings from the study. Authors may like to state reasonable solutions (future studies) as to how to use curcumin PM as an anticancer agent. For example, it might be used in tandem with another natural product or synthetic drug agent.

We have rewritten the section as you suggested: “Curcumin PM nanoparticles had the ability in inhibiting the growth of MCF7 cells and tumor spheroids. However, nanocurcumins were cytotoxic to stromal cells including endothelial cells, fibroblasts, and mesenchymal stem cells. Moreover, nanocurcumin was lethal and teratogenic to zebrafish embryos at the mg/L range. Therefore, we recommend developing nanocurcumin as drugs with reasonable solutions to prevent its side effect on normal cells - for instance, exploring the therapeutical use of nanocurcumin in tandem with other natural or synthetic agents”. (Lines 462 – 468).

Sincerely yours,

Nhung HOANG
